# Amelioration of Functional, Metabolic, and Morphological Deterioration in the Retina following Retinal Detachment by Green Tea Extract

**DOI:** 10.3390/antiox13020235

**Published:** 2024-02-15

**Authors:** Kai On Chu, Yolanda Wong Ying Yip, Kwok Ping Chan, Chi Chiu Wang, Danny Siu Chun Ng, Chi Pui Pang

**Affiliations:** 1Department of Ophthalmology and Visual Sciences, The Chinese University of Hong Kong, Hong Kong, China; chukaion@cuhk.edu.hk (K.O.C.); yolandayip@cuhk.edu.hk (Y.W.Y.Y.); signorickp@cuhk.edu.hk (K.P.C.); 2Department of Obstetrics and Gynaecology, The Chinese University of Hong Kong, Hong Kong, China; ccwang@cuhk.edu.hk

**Keywords:** retinal detachment, green tea extract, apoptosis, oxidative stress, inflammation, metabolism, antioxidant, epigallocatechin gallate

## Abstract

Retinal detachment (RD) can result in the loss of photoreceptors that cause vision impairment and potential blindness. This study explores the protective effects of the oral administration of green tea extract (GTE) in a rat model of RD. Various doses of GTE or epigallocatechin gallate (EGCG), the most active ingredient in green tea catechins, were administered to Sprague Dawley (SD) rats with experimentally induced retinal detachment. The rats received sub-retinal injections of hyaluronic acid (0.1%) to induce RD and were given different doses of GTE and EGCG twice daily for three days. Notably, a low dose of GTE (142.9 mg/kg) caused significantly higher signal amplitudes in electroretinograms (ERGs) compared to higher GTE doses and any doses of EGCG. After administration of a low dose of GTE, the outer nuclear layer thickness, following normalization, of the detached retina reduced to 82.4 ± 8.2% (Mean ± SEM, *p* < 0.05) of the thickness by RD treatment. This thickness was similar to non-RD conditions, at 83.5 ± 4.7% (Mean ± SEM) of the thickness following RD treatment. In addition, the number of TUNEL-positive cells decreased from 76.7 ± 7.4 to 4.7 ± 1.02 (Mean ± SEM, *p* < 0.0001). This reduction was associated with the inhibition of apoptosis through decreased sphingomyelin levels and mitigation of oxidative stress shown by a lowered protein carbonyl level, which may involve suppression of HIF-1α pathways. Furthermore, GTE showed anti-inflammatory effects by reducing inflammatory cytokines and increasing resolving cytokines. In conclusion, low-dose GTE, but not EGCG, significantly alleviated RD-induced apoptosis, oxidative stress, inflammation, and energy insufficiency within a short period and without affecting energy metabolism. These findings suggest the potential of low-dose GTE as a protective agent for the retina in RD.

## 1. Introduction

Retinal detachment (RD) entails the separation of the neurosensory retina from the retinal pigment epithelium, often resulting in irreversible vision loss without immediate surgical intervention. This condition occurs as the detached retina is deprived of essential oxygen and nutrients from the systemic circulation. Central vision loss can result when the macula becomes involved in the extension of RD. The injury to the retina can cause ocular inflammation and structural damage due to proliferative vitreoretinopathy if timely treatment is unavailable, highlighting the urgency of addressing RD promptly.

Individuals experiencing RD typically report sudden flashes of light (photopsia), increased floaters, and the perception of a shadow in their peripheral vision that gradually encroaches toward the center. Surgical intervention is crucial to reattach the retina and minimize damage, as the detached retina often cannot recover without a surgical procedure. However, numerous studies have revealed delays in the surgical process due to various factors. The median interval between the onset of symptoms and surgery ranges from 14 days to 4.2 weeks [1,2]. Patients themselves contribute to this delay, with a median delay of up to five days, further exacerbated by waiting periods for surgery that can extend up to five days. Patient-related delays are often attributed to a lack of awareness regarding RD [1]. Moreover, during the COVID-19 epidemic, surgical delays were prolonged due to limited staffing, operation slots, and the availability of operation theatres [3].

Crucially, surgery should be performed within 72 h to preserve vision effectively [4]. Unfortunately, we lack pharmacological interventions to minimize retinal damage during the critical waiting time. The detached retina, deprived of essential nutrients and oxygen, experiences hypoxia, oxidative stress, inflammation, and apoptotic stress. Green tea extract (GTE) has demonstrated remarkable anti-oxidative, anti-inflammatory, and anti-apoptotic properties studied by different experimental models. Herein, we propose using GTE during the waiting period to mitigate retinal damage and preserve vision. Employing a Sprague Dawley (SD) rat model with induced retinal detachment, we administered GTE orally over three days to assess its potential to minimize damage, compared to a control group and an RD group.

## 2. Materials and Methods

### 2.1. Materials

The green tea extract (GTE), Theaphenon E^®^, kindly provided by Dr. Yukihiko Hara, contained 70% epigallocatechin gallate (EGCG), 5% epigallate catechins (EGC), 4% epicatechin (EC), and 0.6% gallocatechin (GC) [5]. Sodium hyaluronate (1%) (DuoVisc^®^) was obtained from Alcon-Couvreur (Puurs, Belgium). Total protein assay was obtained from Pierce^TM^ BCA Protein Assay Kit (Rockford, IL, USA). The TUNEL assay kit was purchased from ABCAM (Waltham, MA, USA). The Rat Caspase-3 ELISA Kit was obtained from CUSABIO (Houston, TX, USA). The Protein Carbonyl Content Assay Kit was purchased from ABCAM (Waltham, MA, USA). The Rat MCP-1 instant ELISA^TM^ Kits were obtained from Invitrogen^TM^ (Vienna, Austria). The Rat XL Cytokine Array Kit of Proteome Profiler^TM^ Array was purchased from the R&D system (McKinley Place, MN, USA). Fatty acid methyl ester C4–C24 standard mixture and BSTFA+TMCS (99:1) were obtained from Supelco (Bellefonte, PA, USA). Cholesterol, cholesterol-2,3,4-^13^C_3_, and palmitic acid-d_31_ were purchased from Sigma-Aldrich (St. Louis, MO, USA). Decanoic acid-d_19_ was obtained from Cambridge Bioscience (Cambridge, UK). Trimethylsilydiazomethane was purchased from Alfa Aesar (Lancashire, UK). All other reagents were of the highest grade available.

### 2.2. Animal Experiments

All experiments were conducted according to the Association for Research in Vision and Ophthalmology (ARVO) statement on the use of animals. Ethics approval for this study was obtained from the Animal Ethics Committee of the Chinese University of Hong Kong. Female Sprague Dawley rats (about 250 g, 6–8 weeks old) were obtained from the Laboratory Animal Service Centre of the Chinese University of Hong Kong. All animals were housed at 25 °C with 12/12 h light-dark cycles, allowed to access food and water freely, and acclimated for a week. After overnight fasting, the weights of the rats were recorded. The animals were divided into three groups: (a) the non-RD group, a negative control, with a needle slightly pierced into the posterior of the sclera; (b) the RD group with 2 µL of 1% sodium hyaluronate injected into the sub-retinal space; and (c) the GTE treatment group with different doses of GTE fed four hours after RD induction. The animals freely accessed food and water for three days. Meanwhile, the non-RD group and RD group were intragastrically fed with 0.5 mL water twice a day while the GTE group was intragastrically given appropriate doses of GTE twice a day. The doses were determined with reference to previous studies [6,7]. On the fourth day, the rats were anesthetized for ERG examination (Figure 1) or sacrificed by taking blood through a heart puncture and cervical dislocation for histological examination and various assays.

### 2.3. Retinal Detachment Induction

Rats from each group were anesthetized by a mixture of ketamine (75 mg/kg) and xylazine (10 mg/kg) i.p. After cutting the whiskers and eyelashes, disinfecting with iodine solution, and topically anesthetizing with 0.5% proparacaine hydrochloride eye drops to the eye, the pupil was dilated with 5% phenylephrine and 0.5% tropicamide eye drops. The temporal conjunctiva was incised at 5 mm posterior to the limbus and separated from the sclera. For the non-RD group, the tip of a 30 G needle was lightly pierced into the posterior part of the sclera through the incision of the conjunctiva without injection. For the RD group and GTE group, about 2 µL of 1% sodium hyaluronate, DuoVisc^®^ (Alcon-Couvreur, Puurs, Belgium) was injected into the subretinal region through the posterior sclera to induce RD followed by corneal paracentesis with a 30 G needle to normalize the intraocular pressure [8]. Fundus photos were taken by the Eyemera Complete Eye Imaging System by IISCIENCE (San Jose, CA, USA) to demonstrate successful RD induction (Figure 2). Both eyes were induced for each rat. Four hours after the induction, the rats from the non-RD group and RD group were intragastrically (i.g.) fed with 0.5 mL water while GTE or EGCG was fed i.g. in an appropriate dose to the treatment groups. They were fed twice daily for three days.

### 2.4. Histology and Retinal Ganglion Cell Layer Thickness Measurement

The rat eyes were enucleated after three days of treatments. They were fixed in neutral buffered 10% paraformaldehyde overnight. After paraffin embedding, the retina was sectioned from ora serrata to the optic nerve at 5 μm thickness followed by Hematoxylin-Eosin (H&E) staining for the retinal layer measurement. Three rats from each group were used for the test. Two eyes from each rat were enucleated for sectioning. Ten sections from each eye were used to calculate the normalized outer nuclear layer (ONL) thickness. Since the retinal thickness changes with distance from the optic nerve and the retinal thickness varies between individuals, the thickness of the inner nuclear layer (INL) was used to control the ONL measurements [9] (Figure 3a). The ONL thickness was used as an indicator of retinal damage because it has been found that the ONL is disorganized, but there was no significant change in the INL at 7 days after RD [10]. The attached retina was used to further control the detached retina [11]. The thickness of the normalized ONL was used for comparison as follows:Normalized ONL (Nor-ONL) thickness = Average {detached area (ONL/INL) thickness ÷ attached area (ONL/INL) thickness from the same eye}

### 2.5. Electroretinography

Rats from each group were dark-adapted for at least 12 h before electroretinography (ERG). Two batches of rats were tested. The first batch compared the non-RD group, RD, and EGCG treatment group at 96.3 mg/kg and 352.2 mg/kg, and the GTE treatment group at 142.9 mg/kg (contained 96.3 mg/kg EGCG) four hours after RD. The second batch compared non-RD, RD, and GTE treatment groups at 275 mg/kg (contained 185.3 mg/kg EGCG) and 550 mg/kg (contained 370.6 mg/kg EGCG) four hours after RD (Figure 4). The rats were anesthetized by i.p. injection with a ketamine (100 mg/kg)/xylazine (10 mg/kg) mixture. The pupils were dilated with 0.5% tropicamide and 0.5% phenylephrine hydrochloride eye drops, and the eyes were anesthetized with 0.5% *w*/*v* proxymetacaine HCL for ERG examination. The body temperature was kept at 37 °C with a heating pad. ERGs were recorded with a corneal gold wire electrode with a reference electrode at the mouth and grounded in the tail. The amplitude of the b-wave was measured from the trough of the a-wave to the peak of the b-wave in both eyes from each group.

### 2.6. TUNEL Staining

Following trans-cardiac perfusion with 4% paraformaldehyde (PFA), the eyes were enucleated, the cornea punched, and fixed overnight at 4 °C in 4% PFA. The posterior eyecups were cut and the vitreous was removed. The retinae were isolated and stained with TUNEL according to the manufacturer’s instructions for the TUNEL assay kit (ABCAM, Waltham, MA, USA). The retinae were incubated with the staining solution for 1 h at 37 °C and rinsed before being incised into four radial cuts and flat mounted onto microscope slides. The retinae were counterstained with DAPI. TUNEL-positive cells were considered when the green, fluorescent signal co-localized with a DAPI-stained cell nucleus with a brighter spot than the faint green background. TUNEL-positive cells were counted in each field for three fields of each retina randomly around the optic disk under a fluorescence microscope with magnification at 400×. Six retinae were counted each for the non-RD, RD, and GTE treatment groups.

### 2.7. Caspase-3 Assay

After being sacrificed, the eyes of the rats were enucleated and dissected. The retinae were extracted, and the ELISA assay was performed according to the protocol of the Rat Caspase-3 ELISA Kit by the manufacturer (CUSABIO, Houston, TX, USA). In brief, six retinae from each group were rinsed with phosphate buffer, homogenized, and stored overnight at −20 °C. After two freeze-thaw cycles, the homogenates were centrifuged for 5 min at 5000× *g*, 4 °C. The supernatant was taken for ELISA assay and total protein assay by Pierce^TM^ BCA Protein Assay Kit.

### 2.8. Sphingomyelin Assay

Sphingomyelin assay was conducted following the manufacturer’s protocol using the Sphingomyelin Quantification Assay Kit (Sigma-Aldrich^®^, St. Louis, MO, USA). Retinae were homogenized in 0.5 mL of SM Assay Buffer, and the supernatant was obtained by centrifuging the homogenates at 10,000× *g* for 5 min at 4 °C. A 20 µL aliquot of the homogenate was mixed with 20 µL of SM Assay Buffer and heated for 1–2 min at 70 °C until it became cloudy. After centrifugation at 10,000× *g* for 2 min at room temperature, the supernatant was pipetted out. Samples (1–5 µL) were mixed with SM Assay Buffer. Master Reaction Mix (50 µL) was added to each sample and standard control well. The plate was incubated for 60 min at 37 °C, and absorbance at 570 nm was measured.

### 2.9. Protein Carbonyl Content Assay

Six retinae from each group were processed following the protocol of the Protein Carbonyl Content Assay Kit (ABCAM, Waltham, MA, USA). Retinae were homogenized in water and centrifuged. Samples were diluted with water to a concentration of 0.5–2 mg in 100 μL protein per assay. DNPH (100 μL) was added to each sample, vortexed, and incubated for 10 min at room temperature. TCA (30 μL) was mixed with each sample, vortexed, placed on ice for 5 min, and then centrifuged for 2 min. The supernatant was discarded without disturbing the pellet. The pellet was suspended with cold acetone and washed. After sonication to disperse the pellets, they were incubated at −20 °C for 5 min, centrifuged for 2 min, and the acetone was removed. Guanidine solution (200 μL) was added, and the mixture was sonicated to solubilize the proteins. After a brief spin to remove insolubilized material, each sample (100 μL) was transferred to a 96-well plate and measured at approximately 375 nm.

### 2.10. Hypoxia Inducible Factor 1 Alpha (HIF-α) Assay

Six retinae from each group were processed following the protocol of the Rat Hypoxia Inducible Factor 1 Alpha (HIF-1alpha) ELISA Kit (MYBioSource.com, San Diego, CA, USA). Briefly, retinae were rinsed with phosphate buffer, homogenized, and stored overnight at −20 °C. After two freeze-thaw cycles, the homogenates were centrifuged for 5 min at 5000× *g* at 2–8 °C. The supernatant was used for both the ELISA assay and total protein assay using the Pierce^TM^ BCA Protein Assay Kit. For the ELISA assay, the sample and buffer were incubated with HIF-1alpha-HRP conjugate on a pre-coated plate for one hour. After five washes, the wells were incubated with an HRP enzyme substrate, producing a blue-color solution. Adding a stop solution transformed the solution to yellow, and the samples were measured at 450 nm in a microplate reader.

### 2.11. Cytokine Array Analysis

The underlying mechanisms governing the regulation of signaling molecules such as cytokines, chemokines, and growth factors in response to retinal detachment (RD) and green tea extract (GTE) treatment were investigated through proteomic profiling of vitreous humor. The procedure followed the protocol outlined in the Rat XL Cytokine Array Kit of Proteome Profiler^TM^ Array (R&D system, McKinley Place, MN, USA). Vitreous humor samples from two eyes within each group pooled for each membrane in the assay, and three vitreous humor samples from each group were taken for expression comparison.

After blocking the membrane with a blocking buffer for one hour at room temperature, the membrane was incubated with 1.5 mL of a diluted sample solution (80 µL pooled vitreous humor diluted to 1.5 mL) at 4 °C overnight. Subsequently, the membrane was incubated in a detection antibody cocktail for one hour at room temperature, followed by streptavidin-HRP solution for 30 min. Following a washing step, the membrane was immersed in 1 mL Chemi reagent for 1 min, and spot images on the membrane were captured through chemiluminescence detection using the ChemiDoc MP Imaging System (Biorad, Hercules, CA, USA). The optical density of each spot was corrected by the density of the background spot.

Cytokines were grouped based on their major physiological roles associated with inflammation, protection, and metabolic modulation.

### 2.12. Fatty Acids Profiling and Cholesterol Level in Retina

Retinal levels of fatty acids and cholesterol were determined through GC/MS analysis. Ten retinae from each group were homogenized with 200 µL phosphate buffer using repeat pipetting and freeze-thaw cycles. Following centrifugation for 15 min at 4 °C and 13,000 rpm, the supernatant was taken for total protein, fatty acids profiling, and cholesterol assay. For the Bicinchoninic acid (BCA) total protein assay (Pierce^TM^, Rockford, IL, USA), 5 µL of the supernatant was taken.

The remaining supernatant was extracted twice with 200 µL ice-cold Folch solution (chloroform:methanol, 2:1). After spiking with internal standards (arachidonic acid-d_8_, decanoic acid-d8, and decanoic acid-d_9_), the organic solvent was dried by nitrogen purge. The free fatty acids in the residues were selectively derivatized by 50 µL 0.2 M trimethylsilyl diazomethane (TMSD) in a methanol/acetone (1:9) mixture [12,13]. A 1 µL solution was pipetted out for cholesterol assay, and the remaining solution was incubated at room temperature for 30 min. The solution (1 µL) was injected into GC/MS for methylated fatty acid profiling analysis.

For cholesterol determination, a split 1 µL solution was diluted 100-fold with dichloromethane. Then, 50 µL of the solution was spiked with the internal standard cholesterol-^13^C_3_ at 10 µg/mL. After nitrogen purge to dryness, 50 µL of BSTFA:pyridine (1:2) mixture was added and derivatized for 30 min at 60 °C. Following nitrogen purge drying, the residues were dissolved in 50 µL dichloromethane, and 1 µL of the solution was injected into GC/MS for cholesterol analysis [14].

### 2.13. GC/MS Analysis

Fatty acids were analyzed using an Agilent 6890/5973 MSD gas chromatography mass spectrometer. Methylated fatty acids were separated using an HP-88 60 m × 0.25 mm × 0.2 µm capillary column (Agilent, Santa Clara, CA, USA). The temperature program was set as follows: 80 °C for 5 min, 200 °C with a ramp of 4 °C/min, 240 °C with a ramp of 2 °C/min and held at 240 °C for 10 min. The inlet temperature was 250 °C with splitless mode for 2 min. The MSD operated with an electron multiplier (EM) voltage of 1600 V. The level of fatty acids was determined by selected ion monitoring (SIM) mode by selecting characteristic fragments. Identification was made according to the retention time and qualifying ions according to their corresponding standard methylated fatty acids. Concentrations were normalized by the total protein in each retina, quantified against internal standards, and compared to calibration curves derived from different concentrations of standard methylated fatty acid mixtures.

Cholesterols were determined under similar parameters to methylated fatty acids, except for the temperature program: 160 °C for 5 min, 200 °C with a ramp of 4 °C/min, 200 °C for 0 min, 240 °C with a ramp of 2 °C/min and held at 240 °C for 10 min. Quantification was performed referencing the cholesterol-^13^C_3_ internal standard and utilizing a standard calibration curve constructed from various concentrations of standard solutions.

The concentration of each fatty acid in each treatment group was compared to the non-RD group, and fold changes in the fatty acids of each group were determined.

### 2.14. Statistical Analysis

For comparison between pairs of groups, the Mann–Whitney test or *t*-test were used for calculation where appropriate. For comparison between all groups, ANOVA with Tukey post hoc test was used.

## 3. Results

### 3.1. Retinal Detachment Features

The retina detached after each hyaluronic acid injection (Figure 2). The retinal detachment remained for three days until the day of sacrifice. Minor bleeding associated with the retinal detachment appeared after the injection.

### 3.2. Comparison of Retina Thickness under Different Treatments

The detached retina’s swelling after the RD lesion was shown by the thickness of normalized ONL of the detached retina, which was thicker than the non-RD and the other treatment group (Figure 3 and Appendix A). The average normalized ONL thickness of the detached retina from the GTE group at 142.9 and 550 mg/kg was significantly thinner than the RD group with 82.4 ± 8.2% (Mean ± SEM, *p* < 0.05) and 66.0 ± 8.2% (Mean ± SEM, *p* < 0.05), respectively, compared to the thickness in RD from the two batches. The thickness at a low dose of GTE was similar to that in non-RD conditions with 83.5 ± 4.7% (Mean ± SEM). The other treatments did not show significant changes, although they appeared thinner than the RD group (*p* < 0.05, n = 6).

### 3.3. Comparison of Electroretinography Responses under Different Treatments

The amplitudes of the a-wave and b-wave of the non-RD and GTE at dose 142.9 mg/kg were significantly higher than the RD group, although the amplitude of the b-wave of the low dose GTE group was still lower than the non-RD group. However, further increases in the doses of GTE to 275 mg/kg and 250 mg/kg lowered the amplitudes of the a- and b-waves even below the RD group. The amplitudes of the a-wave and b-wave of the two doses of EGCG groups, 96.2 and 352.2 mg/kg, showed no significant difference from the RD group. (Figure 4 and Appendix A).

### 3.4. Comparison of the Number of Apoptosis Cells under Different Treatments by TUNEL Assay

The number of TUNEL-positive cells largely increased in the retina after RD. The number of TUNEL-positive cells was greatly reduced after administration of a low dose of GTE (142.9 mg/kg) twice daily for three days. 

### 3.5. Comparison of the Expression of Pro-Apoptotic Caspase-3, Sphingomyelin, Protein Carbonyl, and HIF-1α Levels in the Retinae under Different Treatments

The pro-apoptotic caspase-3 level in the retina after the detachment significantly increased and did not significantly decrease after the administration of GTE. The sphingomyelin level of the retinae significantly decreased after administration of a low dose of GTE. The oxidative stress significantly decreased after administration of GTE, as indicated by the protein carbonyl level decrease. There was no significant difference in oxidative stress between the non-RD and RD groups. The inflammation level of the RD group was significantly higher than the other group, as indicated by the significantly higher level of HIF-1α present in the retina. The inflammation of the retina was relieved after GTE administration.

### 3.6. Comparing the Cytokines Proteome Profile in the Vitreous Humor under Different Treatments by Cytokine Array Analysis

Some cytokines differentially expressed in the vitreous humor are associated with inflammation, cytoprotection, and metabolism regulation. Amongst the cytokines, the levels of CCL2, CCL20, GDF, OPN, MMP3, serpin E1, adiponectin, DPPIV, IGF-1, and IGFBP-3 in the RD group increased more than 2.5-fold compared to non-RD group, whereas IL-1α and IL-13 in the GTE group increased more than 2.5-fold compared to the other groups.

### 3.7. Comparing the Fatty Acids Profile in the Retinae under Different Treatments by GC/MS Analysis

Free fatty acids significantly increased over 30-fold after RD compared to the non-RD group. There was no significant difference in the level of fatty acids between the RD and GTE treatment group.

### 3.8. Comparing the Cholesterol Level in the Retinae under Different Treatments by GC/MS Analysis

There was no significant difference in cholesterol levels between the three groups.

## 4. Discussion

The therapeutic benefits of green tea have been attributed to its antioxidative, anti-apoptotic, and anti-inflammatory properties [15,16,17,18]. Experimental models of ocular diseases, including ocular inflammation, retinal and choroidal neovascularization, optic nerve damage, and cataracts, have shown improved outcomes with green tea extract, encompassing flavanols and catechins [19,20,21,22]. Nerve cells are prone to damage following retinal detachment (RD) due to oxidative stress, inflammation, and nutrient deprivation [23,24]. To the best of our knowledge, this is the first report on minimizing nerve cell damage by green tea extract after retinal detachment.

In our prior studies, we observed varying biological effects with different dosages of green tea extract (GTE) and catechins [17,25]. Despite epigallocatechin gallate (EGCG) having been demonstrated as a dominant contributor to the beneficial effects of GTE [26], our findings indicated that EGCG alone lacks a significant anti-inflammatory effect in ocular inflammatory models, such as experimental uveitis (EIU) and experimental autoimmune uveitis (EAU), compared to GTE containing an equivalent EGCG dosage [6,7]. On the other hand, we observed increased pro-oxidative effects with elevated EGCG content in GTE in vivo [17]. In a metabolomics study, EGCG exhibited more detrimental effects on human umbilical vein endothelial cell (HUVEC) growth when compared to GTE [25].

In this study, we studied different doses of GTE and their corresponding EGCG doses following retinal detachment induction (Figure 2) to assess the potential of green tea in delaying retinal damage (Figure 1). Consistent with our previous findings, both high and low doses of EGCG exhibited detrimental effects on the detached retina in this experimental model. However, our results show that a low dose of GTE (142.9 mg/kg) exerted protective effects, as evidenced by reductions in retinal edema indicated by normalized outer nuclear layer (ONL) thickness and functional electroretinogram (ERG) tests (Figure 3 and Figure 4). Specifically, the group given a low dose showed significantly reduced retinal edema. The amplitude of the a-wave and b-wave in the scotopic test, and the b-wave amplitude in the photopic test, were significantly higher than in the retinal detachment group. The amplitude of the a-wave in the scotopic test and b-wave in the photopic test from the low-dose group showed a similar magnitude to the non-retinal detachment group. Due to its superior functional protective effects, we selected a low dose of GTE (142.9 mg/kg) for further investigation.

The administration of a low dose of green tea extract (GTE) significantly reduced the number of TUNEL-positive cells, indicating protection against neuron cell death (Figure 5). Following retinal detachment, apoptosis, necrosis, and necroptosis contribute to retinal cell death [27,28]. Persistently elevated caspase-3 levels in the GTE-treated retina appeared (Figure 6), suggesting a protective factor against cell death in mildly stressed cells [29]. This protection may be attributed to the activation of the anti-apoptotic Akt kinase, subsequently stimulating the mTOR [30] and NF-κB pathways [31] during inflammation. Activated Akt also inhibits pro-apoptotic molecules such as Bad [32], supporting the notion that elevated levels of caspase-3 protect cells from apoptosis.

Consistent with our previous study [19], a low dose of GTE had an anti-oxidative effect that significantly reduced the carbonyl level in the retina, which supports the idea that GTE exerted strong anti-oxidation to suppress the oxidative stress of the retina even under hypoxic conditions (Figure 6 and Appendix A). The relief of oxidative stress was also demonstrated by the decreased HIF-1α expression (Figure 6). HIF-1α is an oxygen-sensitive master regulator of numerous hypoxia-inducible genes for hypoxia. It controls genes that regulate angiogenesis, cell proliferation/survival, and glucose/iron metabolism. It diverts pyruvate away from the tricarboxylic acid (TCA) cycle and oxidative phosphorylation (OXPHOS) in the mitochondria in oxygen-deprived cells to a less energy-efficient metabolism of pyruvate to lactic acid. Therefore, it leads to cellular adaptation to hypoxia to enhance glucose uptake, and production of lactate, and to reduce mitochondrial respiration [33]. GTE is known to inhibit hypoxia-induced HIF-1α protein accumulation in cancer cells by blocking the phosphatidylinositol 3-kinase/Akt and extracellular signal-regulated kinase 1/2 signaling pathways and elevating the protein degradation without affecting the mRNA expression [34].

Sphingomyelin is a vital constituent of plasma membrane lipids essential for myelin sheath formation in nerve axons. It constitutes membrane microdomains, such as lipid rafts, to regulate trans-membrane signaling and affects cell migration, apoptosis, autophagy, and cell survival/proliferation [35]. Retinal cells exhibit increased resistance to apoptosis at lower sphingomyelin levels. For instance, apoptosis-resistant S49 mouse lymphoma cells have lower sphingomyelin levels than their chemo-sensitive counterparts [36]. The significantly elevated level of pro-apoptotic caspase-3 and reduced level of sphingomyelin in the retinal detachment group compared to the non-retinal detachment group (Figure 6) suggest that GTE prevents cells from apoptosis by lowering the sphingomyelin level, even in the presence of activated pro-apoptotic mechanisms. As a result, the number of TUNEL-positive cells significantly decreased following GTE treatment.

In our investigation of the chemokine profile in the extracellular vitreous humor using a high-sensitive chemiluminescence detection system, we observed a significant increase in pro-inflammatory cytokines (CCL2 and CCL20), the macrophage inhibitory cytokine GDF-15, anti-apoptotic factors (osteopontin—OPN), matrix metalloproteinase (MMP3), urokinase plasminogen activator (uPA) inhibitor (serpin E1), and metabolism regulators (adiponectin, DPPIV, IGF-1, and IGFBP-3) following retinal detachment (Figure 7, Appendix A). GTE treatment led to a significant reduction in the levels of CCL2, serpin E1, adiponectin, DPPIV, IGF-1, and IGFBP-3, while the CCL20, GDF-15, OPN, and MMP-3 levels’ reduction was insignificant. These results suggest that retinal detachment activated the innate immune response, recruiting monocytes, macrophages, lymphocytes, and dendritic cells, as indicated by the upregulation of CCL2 and CCL20 [37,38]. Furthermore, the homeostatic regulatory immune inhibition factor GDF-15, a macrophage inhibitory cytokine, is activated after retinal detachment [39]. Following retinal detachment, GDF-15 induces apoptosis by activating caspase-3. GTE is able to suppress GDF-15. It plays a protective role in the eyes after retinal detachment.

Following the induction of retinal detachment, osteopontin (OPN), an anti-apoptotic factor, is released. OPN prevents non-programmed cell death during inflammation [40]. OPN also induces the function of Th1 cytokines and immunoglobulin production during inflammation. We found that GTE treatment did not significantly suppress OPN production. Therefore, it did not influence the adaptive immune response and activate the retinal cells to the apoptosis pathway.

Matrix metalloproteinase, MMP-3, was upregulated following retinal detachment. MMP-3 can repair wounds by remodeling connective tissues by breaking down extracellular matrices such as collagen, proteoglycans, and fibronectin. GTE did not affect this activity [41]. However, retinal detachment caused an increase in serpin, an inhibitor of uPA, an enzyme for plasmin formation [42]. Plasmin degrades extracellular matrices and blood clots. Hence, GTE treatment promoted extracellular matrix modulation and prevented blood clot formation.

An important finding regarding the efficacy of GTE in maintaining retinal cell survival in this study was the metabolic effects. Retinal detachment causes the retinal cells to be deprived of nutrients by the blocking of supplies from the retinal epithelial layer and choroid tissues. Adiponectin releases enhanced insulin sensitivity to promote the influx of glucose and increase fatty acid oxidation through the elevation of phosphorylation of acetyl Co-A carboxylase to meet the intense energy demand [43]. DPP4 also affects glucose metabolism and insulin sensitivity and causes local inflammation. It acts as an enzyme to digest glucagon-like peptide-1, by interacting with the extracellular matrix and caveolin-1-associated insulin transduction pathway. As a result, it causes extracellular hyperglycemia and insulin resistance [44]. The increase in DPP4 during retinal detachment thus depresses glucose metabolism in retinal cells. GTE treatment lowered the adiponectin and DPP4.

IGF-1 is essential for the regulation of glucose and lipid metabolism. It increases lipolysis, the transport of fatty acids into tissue, free fatty acid utilization, and lipid oxidation. Furthermore, it enhances glucose influx through insulin receptors [45]. The elevation of IGF-1 level during retinal detachment is a homeostatic response to diverting energy utilization from the external glucose supply to the internal supply through lipid oxidation. We found the level of free fatty acids increased significantly in the retina following the detachment (Figure 8). The source of fatty acids may either come from the breakdown of membranes from the neuron cells or internal lipid deposits. GTE may relieve the energy demand and metabolic stress [7], so the IGF-1 level decreases. Insulin-like growth factor-binding protein 3 (IGFBP-3) interacts and transports IGF-1 in the circulation [46]. Its level increased following retinal detachment but lowered after GTE treatment. Hence, the observed attenuation of lipid metabolism following GTE treatment in retinal detachment indicates a protective effect of GTE from deprivation of energy supply.

In addition, cholesterol, a crucial constituent of the plasma membrane for signaling and maintaining the integrity of the plasma membrane in the retina [47], did not show significant differences among the three groups (Figure 9). Therefore, it showed the membrane integrity of the photoreceptor is likely maintained for three days after retinal detachment. The starvation caused the metabolism to transit from glucose to lipid metabolism, and fatty acids, unsaturated or saturated fatty acids, were released from lipolysis of lipid deposits in the retina [48]. Although necrosis and necroptosis, involving cell membrane disruption, have been reported [9,10], we found insignificant differences in cholesterol levels within the groups, indicating the cell membranes of the two groups remained intact.

IL-13 has been reported to enhance microglial/macrophage anti-inflammatory responses and reduce brain cell death due to ischemia. It helps improve sensory function during cerebral ischemia [49]. The significant increase in IL-13 following GTE treatment supports its anti-inflammatory effect and protective role in the detached retina. IL-1 protects the disrupted mucosal layer during Helicobacter pylori infection [50] because IL-1 signaling increases the tight junction and permeability of the gut epithelium [51,52]. Similarly, the increased IL-1 after GTE treatment in the retina facilitates the signaling pathway to promote nutrient absorption after retinal detachment.

GTE treatment also contributed to the restoration of interleukin cytokines to physiological levels during the recovery from retinal detachment. Following retinal detachment, the interleukin cytokines were depressed for three days after the experimental induction of retinal detachment but recovered after GTE treatment.

This dose is equivalent to approximately 0.984 g of EGCG equivalent dose twice daily for humans after allometric conversion [53]. It falls below a dose of the 2 g EGCG equivalent dose twice daily in the phase II clinical trial, where patients with chronic lymphocytic leukemia tolerated it well [54].

## 5. Conclusions

Our results show effective protection of the retina within three days of retinal detachment by a low dose of GTE (142.9 mg/kg). The protection was evidenced by the increased a- and b-wave amplitudes in the electroretinogram (ERG), the prevention of thickening of the outer nuclear layer (ONL) in histology, and a significant reduction in the number of TUNEL-positive cells. Retinal detachment is associated with inflammation, oxidative stress, and starvation, leading to apoptosis. Consequently, levels of CCL2, carbonyl content, HIF-1α, TUNEL-positive cells, and caspase-3 increased.

Because of nutrient deprivation, retinal cells utilize stored lipid reserves as an energy source instead of glucose through interactions involving DDP4, adiponectin, and IGF-1. The retina responds homeostatically to protect cells by increasing the expression of cytokines, including osteopontin (OPN), matrix metalloproteinase-3 (MMP-3), and serpin. GTE neutralizes free radicals, suppresses the accumulation of HIF-1α, prevents cell death by reducing sphingomyelin, and exerts an anti-inflammatory effect by suppressing inflammatory cytokines such as CCL2 while increasing inflammatory-resolving cytokines such as IL-13. GTE may alleviate the energy demand, which restores the IGF-1, IGFBP-3, DPP4, and adiponectin levels.

Furthermore, GTE treatment helps maintain the utilization of fatty acids, as depicted in Figure 10. Therefore, our study has shown that a low dose of GTE serves as a remedy to retard retinal cell damage before surgical reattachment.

## Figures and Tables

**Figure 1 antioxidants-13-00235-f001:**
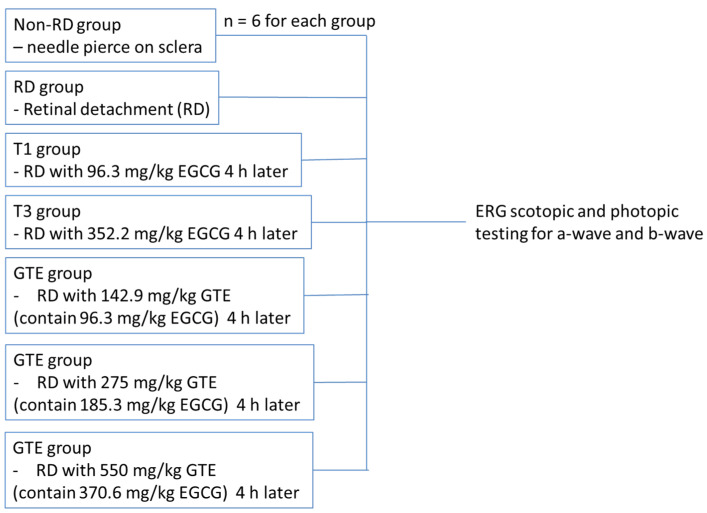
The strategy of ERG testing for non-RD, RD, and different treatment groups.

**Figure 2 antioxidants-13-00235-f002:**
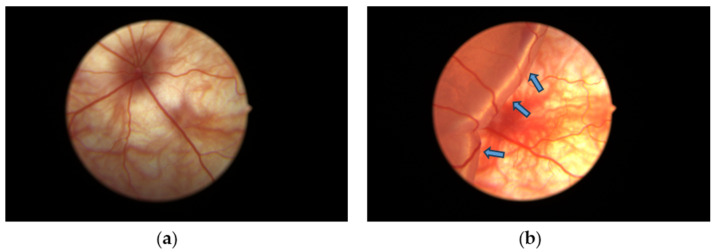
Diagrams showing (**a**) the fundus of the non-RD eye and (**b**) experimental retinal detachment eyes with large edema formed from the detached retina. Blue arrows indicate the swollen detached retina containing subretinal sodium hyaluronate solution introduced after subretinal injection.

**Figure 3 antioxidants-13-00235-f003:**
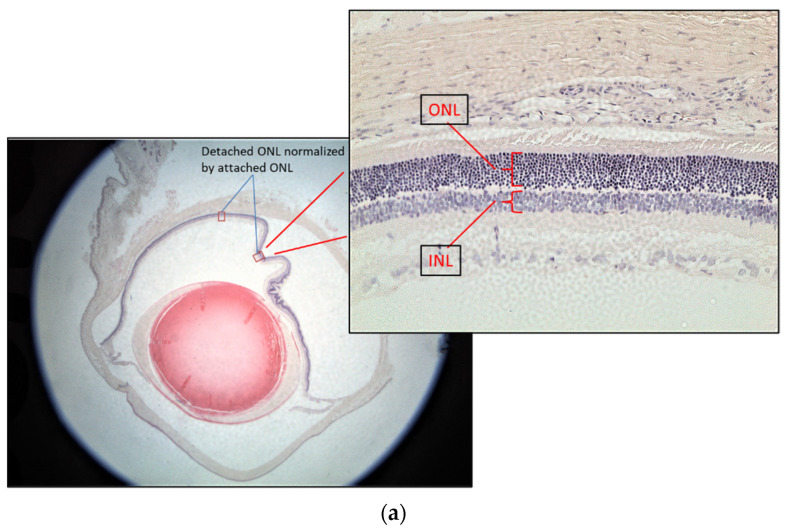
Representative diagrams of retina thickness under different treatments. (**a**) The inner nuclear layer (INL) and outer nuclear layer (ONL), and the retina thickness from the detached and attached segment were used for the calculation; (**b**) comparison of the normalized ONL (Nor-(ONL/INL) under different treatments to the normalized ONL of RD in two testing batch samples. (* denotes significant difference comparing to RD treatment with each treatment (*p* < 0.05, n = 6)). Error bar: - ± SEM.

**Figure 4 antioxidants-13-00235-f004:**
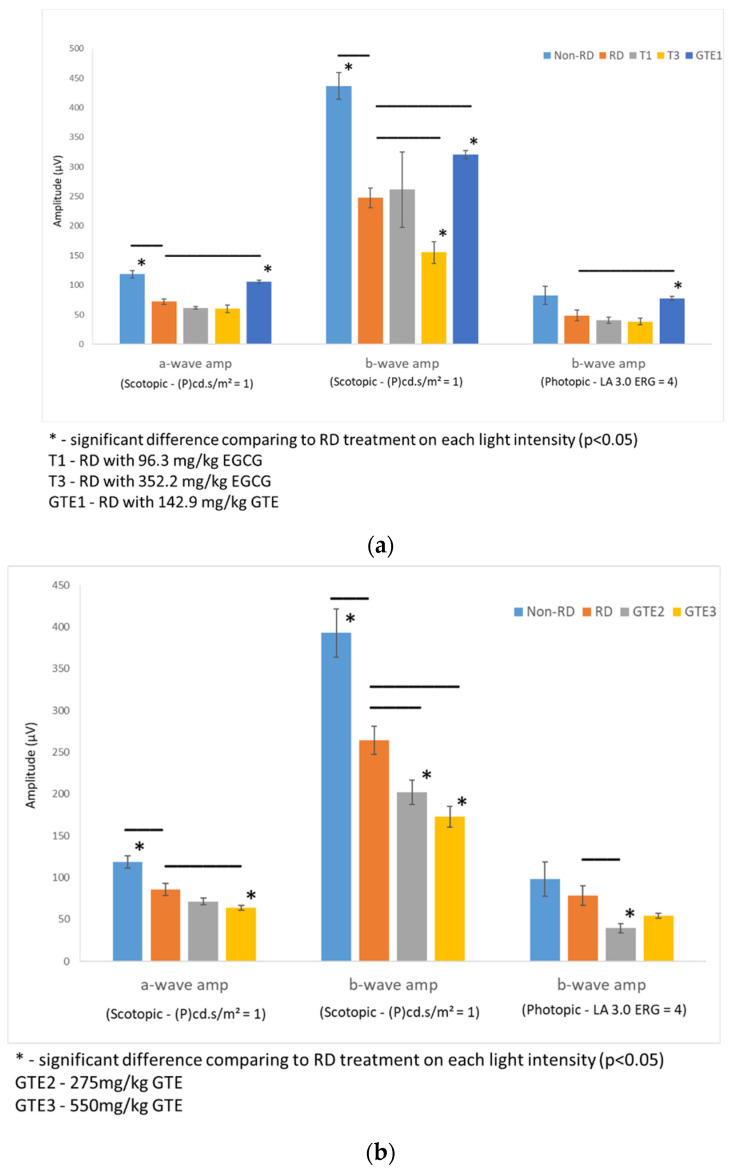
Comparison of retinal responses by ERG scotopic and photopic tests in different light intensities under different treatments. (**a**) The first batch of samples compared a- and b-wave amplitude by an ERG scotopic test, and b-wave amplitude by an ERG photopic test of RD samples to non-RD, low dose (142.9 mg/kg) GTE, low dose (96.3 mg/kg) and high dose (352.2 mg/kg) EGCG treatment groups. (**b**) Second batch of samples compared a- and b-wave amplitude by an ERG scotopic test, and b-wave amplitude by an ERG photopic test of RD samples to non-RD, GTE doses at 275 mg/kg and 550 mg/kg treatment groups. (n = 3 per group). *—The test group was significantly different from the RD group (*p* < 0.05). Error bar: - ± SEM.

**Figure 5 antioxidants-13-00235-f005:**
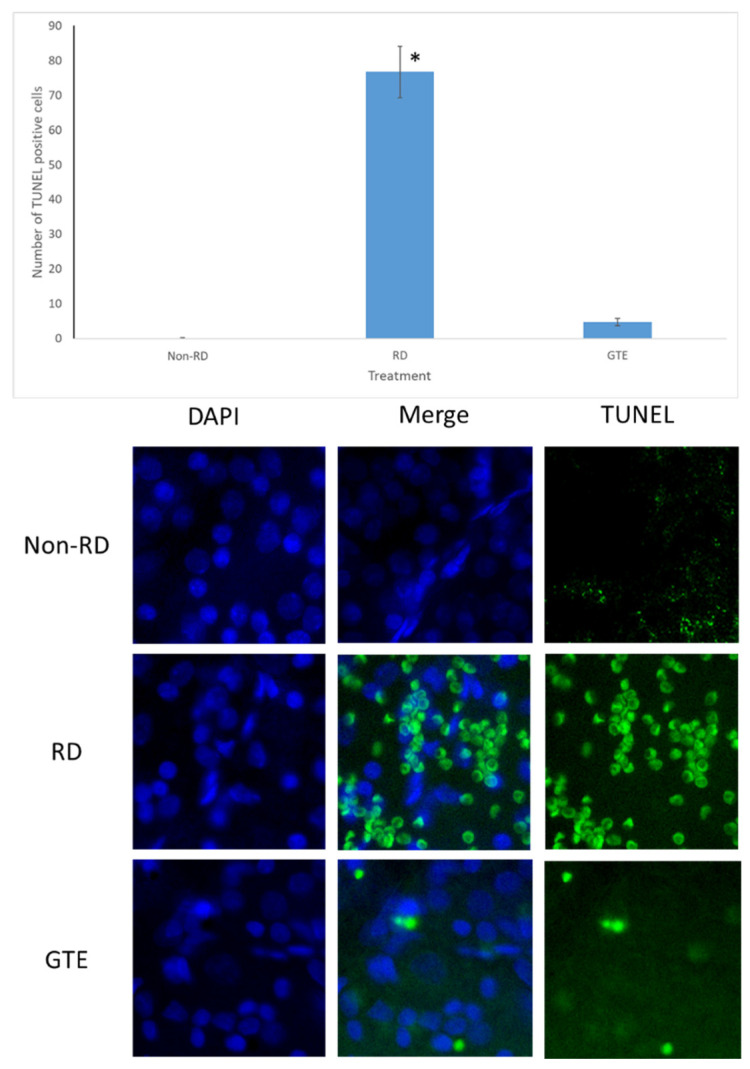
TUNEL test after RD. The representative diagram shows the number of TUNEL-positive cells increased three days after RD but decreased with GTE treatment. The number of TUNEL-positive cells significantly increased in the RD group compared to the non-RD group but significantly decreased compared to the GTE treatment group (142.9 mg/kg). *—significant difference (*p* < 0.0001, n = 6). Error bar: - ± SEM. Magnification 10 × 40.

**Figure 6 antioxidants-13-00235-f006:**
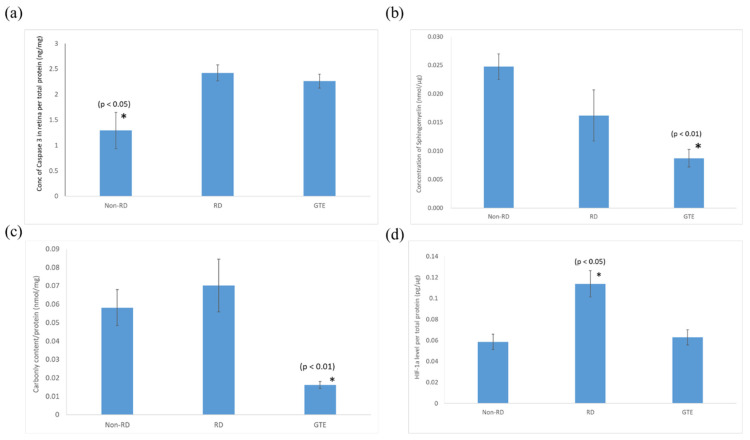
Comparison of (**a**) caspase-3, (**b**) sphingomyelin, (**c**) protein carbonyl, and (**d**) HIF-1α levels in the retina in non-RD, RD, and GTE treatment groups (142.9 mg/kg). *—indicates a significant difference compared to the other groups (*p* < 0.05, n = 6). Error bar: - ± SEM.

**Figure 7 antioxidants-13-00235-f007:**
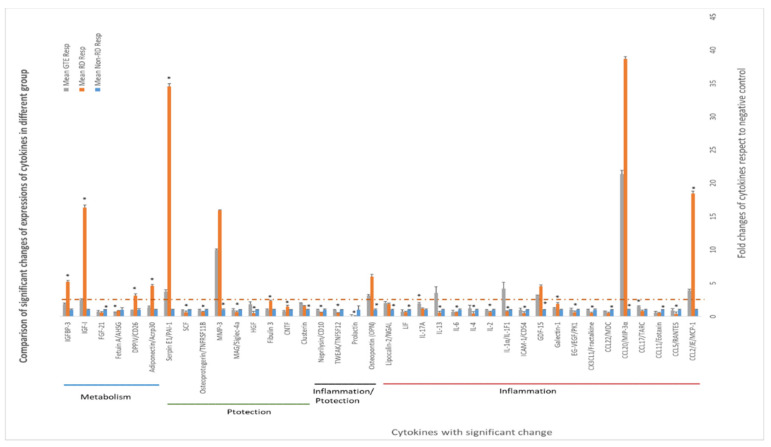
Differential expression of cytokines in the vitreous humors of different study groups. The expression of cytokines in the non-RD group was assigned as 1; the folds of changes in the expression of cytokines in the other groups were compared. ANOVA with post hoc Tukey and Dunnett’s test was used for the comparison. *—significant difference of expressions from the other groups (*p* < 0.05, n = 3). Error bar: - ± SEM. The dashed line indicates over 2.5-fold of changes compared to the non-RD group.

**Figure 8 antioxidants-13-00235-f008:**
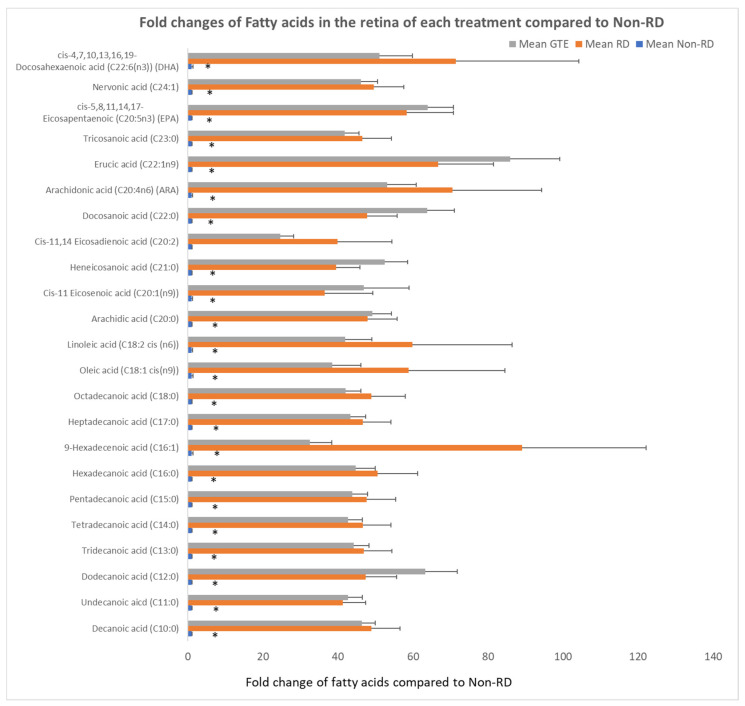
Comparison of fold changes of fatty acid levels of the retina in each group to the non-RD group. The levels of fatty acid were normalized by total protein in each retina. * denotes a significant difference in the levels of fatty acids amongst the groups. Error bar: - ± SEM. *: (*p* < 0.05, n = 10).

**Figure 9 antioxidants-13-00235-f009:**
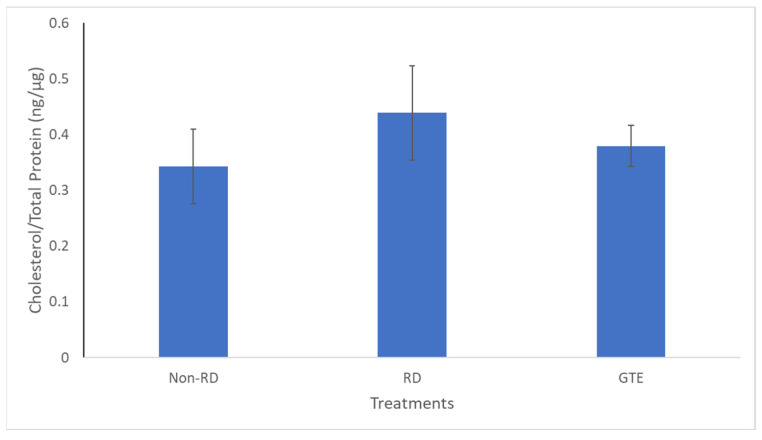
Comparison of the cholesterol level in the retinae between the non-RD, RD, and low-dose GTE treatment groups. The cholesterol level was normalized by total protein in each retina. There was no significant difference between the groups. Error bar: - ± SEM.

**Figure 10 antioxidants-13-00235-f010:**
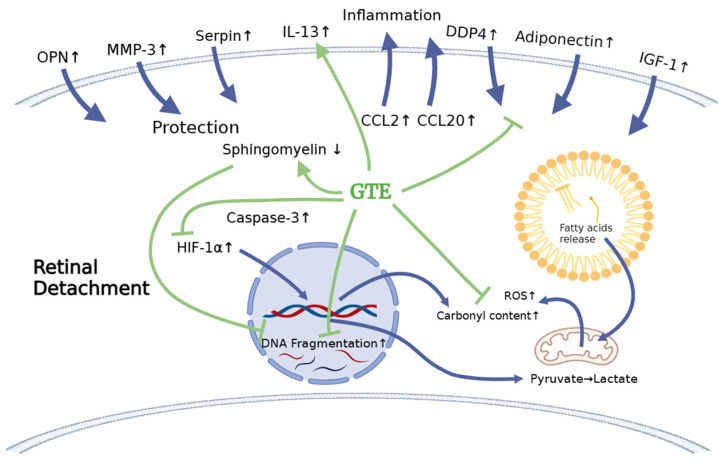
The schematic diagram shows the effects of retinal detachment on the endogenous molecule expression that explains the subsequent physiological effects, including the increase in oxidation, inflammation, and apoptosis. It delineates how the treatment with green tea extract relieves stress. The blue arrows indicate the production of cytokines and metabolites and their infiltration after RD induction. The green arrows indicate the effects of GTE on cytokines after GTE treatment. The green blunt arrows indicate the inhibition effects of GTE on cytokines and metabolite production.

## Data Availability

The original contributions presented in the study are included in the article and Appendix A. Further inquiries can be directed to the corresponding authors.

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
