# Peer review of "Amelioration of Functional, Metabolic, and Morphological Deterioration in the Retina following Retinal Detachment by Green Tea Extract"

_antioxidants, 2024, doi:10.3390/antiox13020235_

Round 1

Reviewer 1 Report

Comments and Suggestions for Authors

Thanks for offering the opportunity to review this paper. There are certain issues with this paper that need to be addressed, so I would like to review it again after the author carefully modifies it.

1. Language: On the whole, I can understand what the authors are trying to express in this manuscript, 9but there are some places where the expressions are not smooth. In addition, there are some grammatical errors or clerical errors in the manuscript.

2. Is seems that some figures can be merged into one without having them to be presented separately.

3. The dose of GTE (EGCG) used by the author in this article is very high. At present, it is impossible to apply this dose to human beings. The author should discuss this point.

4. I think Figure 13 should be beautified.

5. Figure 2 "More than 50 % of retina were detached." I can't judge the are of retinal detachment from the Figure.

Comments on the Quality of English Language

Moderate editing of English language required

Reviewer 2 Report

Comments and Suggestions for Authors

The manuscript is well written and easy to read. The experiments are convincing and support the conclusions drawn. However, there are some concerns that should be addressed, especially concerning the statistical/error bars of the figures:

Table 1 is useless. Information is available on the manufacturer’s site.

Figure 4, why only panel C has error bars. They should be added to every panel.

Some figures are of poor quality, especially the graph and should be significantly improved.

Figure 10, please add errors bars and add a representative image of the blot, at least in supplementary materials.

Figure 11, please add errors bars.

Figure 13 is not useful. I would remove it.

I am not convinced about the statistical significance in figures 3, 4, 6, 7, 9 and 10. The error bars are large and the difference between conditions is not that important. It is difficult to believe that it the difference is significant. It is even a bigger problem in the figures missing error bars (Fig 4, 10). Please provide some raw data analysis to be convinced that the difference is significant.

Reviewer 3 Report

Comments and Suggestions for Authors

Congratulations for a very detailed experiment. The contents are highly detailed, and there seems no major issues. I understand the usefulness of low dose GTE to prevent retinal damage induced by the retinal detachment. However, before evaluating the content of this paper, I felt that there are significant issues with the way the paper is written.

 Minor criticisms:

I recommend to revise the title. The meaning of “deterioration of retina” is unclear. Does it mean morphological change or functional change, or both? The ward of “following detachment” is unclear. What is detached?

Line 41. “The retinal injury causes ocular inflammation and structural damages due to proliferative vitreoretinopathy if not treated on time. Please revise “retina injury”. 

Line 45. Basically. RD resolves spontaneously in rare case.

Line 46. Need revision.

Is the description of reference number correct? Usually, Arabic figure is used.

Line 87 Need revision

Line 105 Where is posterior limbus? Inferior??

Figure 2a The retina appears to be cloudy. Why?

Figure 2b The term “retinal bubble” is strange. The ward is not used to express RD in Ophthalmology.

Line 122 How many days after did the eyeball undergo enucleation?

Comments on the Quality of English Language

There is a problem with the way the information is presented, making it challenging to read. Please review the entire paper and reformat it to adhere to the standard structure of a medical research paper. I recommend seeking the assistance of a native English editor for proofreading and correction of the language and expression.

Round 2

Reviewer 1 Report

The paper can be accepted now.

The paper can be accepted now.

Author Response

The English has been revised.

Reviewer 2 Report

The authors answered adequately to all the concerns I raised. The manuscript can be published in the present form

The authors answered adequately to all the concerns I raised. The manuscript can be published in the present form

Author Response

Thanks for your invaluable comments!

Reviewer 3 Report

Most of the previous points have been addressed but there are still some unusual or wrong expressions.

Line 24 "the normalized outer nuclear layer thickness of the detached retina" I cannot understand the meaning of this phrase.

Line 131. Figure 2. "Edema" is used to represent the intraretinal fluid. I guess this lesion has subretinal fluid. Blue arrows seems to indicate the boundary of retinal detachment.
